

# Optimising AI writing assessment using feedback and knowledge graph integration

Ci Zhang

Wenzhou University, Wenzhou, China

## ABSTRACT

In this work, the authors provide a novel framework for the effectiveness of AI writing assessment systems by embedding state-of-the-art deep learning networks, user feedback mechanisms, and knowledge graph frameworks. Most writing assessment tools cannot give personalized, detailed feedback. To tackle this problem, we employ writing assessment transformer models BERT and GPT-3, which allow exploring and scoring the writing on various features, including phrase structure, semantics, vocabulary usage, *etc*. In our system, we propose a dynamic relational knowledge graph that incorporates writing concepts and their relations, making it easier for the system to devise contextualized thesaurus-wise suggestions. The addition of graph neural networks (GNNs) empowers the model by boosting the GNN's learning ability regarding the knowledge graph and improving comprehension of complex semantics. Additionally, we have included an iterative design whereby user feedback is collected, and the system adjusts the feedback given in light of historical feedback and changes in a user's writing behavior over time. The system reconceptualizes the problem of user AI interaction by incorporating its dynamic nature and movement towards the known user and not *vice-versa*, achieving higher efficiency. To assess user satisfaction and improvements in the quality of the prepared texts, the authors conduct a series of user studies evaluating the efficiency of this integrated system. However, the preliminary data obtained from the task performance analysis show that the results of the proposed framework are far better than those of traditional methods, achieving a better level of engagement and feedback while performing the assessment. This study underscores the potential of deep learning, feedback, and knowledge graph integration in leveraging writing education. It can potentially reform learners' capabilities, enabling them to write better and more effectively.

## INTRODUCTION

Both academic and business communities have long been interested in measuring intellectual progress. To evaluate progress, intelligent educational services like intelligent tutor systems (ITS), intellectual resource recovery, and faculty career assessments, thoroughly evaluating a scholar *via* a systemic review has become necessary (*Tang, 2016*; *Tarus, Niu & Mustafa, 2018*; *Mao et al., 2019*). To find an answer to this issue, several studies have attempted to quantify and qualitatively assess academic performance in areas

Corresponding author
Ci Zhang,
cizhangoptimize@outlook.com,
00061092@wzu.edu.cn

such as research projects (*Gao et al., 2019*), patents (*Faria et al., 2018*), teaching quality (*Ory, 2000*; *Boswell, 2016*), *etc.* Now that the Internet is fully embedded in the academic world, data on intellectual pursuits can be found and shared online.

Intelligent education has undergone a profound transformation—particularly in English language learning—driven by the integration of advanced technologies such as knowledge graphs and large-scale language models (*Zhang & Wang, 2020*). Knowledge graphs are a type of structured data representation where entities (such as concepts, objects, or events) are represented as nodes, and their interrelations are depicted as edges. This graph-based approach captures complex relationships between concepts, making them ideal for tasks requiring semantic understanding. In this study, knowledge graphs help contextualise various writing concepts and their relationships, enabling the system to provide more nuanced, relevant feedback.

In addition, large-scale language models such as Bidirectional Encoder Representations from Transformers (BERT) and Generative Pretrained Transformer 3 (GPT-3) have revolutionised the field of natural language processing (NLP). BERT, introduced by *Ouyang & Hou (2024)*, is a transformer-based model that improves language understanding by processing text bidirectionally. Unlike traditional models that only consider the sequence of words from left to right, BERT looks at both the preceding and succeeding words in a sentence, allowing it to grasp deeper context and better understand the nuances of language. GPT-3, on the other hand, is a generative model built on autoregressive principles, trained to predict the next word in a sequence. With 175 billion parameters, GPT-3 can generate human-like text, making it highly effective for tasks such as text completion, summarisation, and feedback generation.

The integration of these technologies allows for more personalised and dynamic writing assessments. The feedback loop in the system plays a critical role in improving the overall quality of user-generated text. Feedback loops, often utilised in educational systems, involve providing users with suggestions and corrections that help them improve over time. Our framework gives this feedback based on predefined metrics and adjusts it based on user feedback mechanisms that evolve as users interact with the system. Furthermore, to enhance the system's ability to handle complex relationships and contextualize writing suggestions, we incorporate graph neural networks (GNNs). GNNs are a class of neural networks designed to process graph-structured data, allowing them to learn from the relationships between nodes in a graph. Using GNNs, our system can leverage the connections between various writing-related concepts, such as style, grammar, and coherence, enabling it to generate feedback that is deeply aware of the context in which writing occurs.

This study aims to integrate these advanced models—BERT, GPT-3, and GNNs—into a unified framework for AI-driven writing assessments. This framework uses deep learning technologies to provide personalized feedback based on writing quality and adapts to the learner's unique needs, improving over time through iterative interactions. The article also explores the role of user feedback, a critical component in ensuring that the system continues to evolve in line with the user's learning style and progression. This study aims to address this gap by exploring the following research question: How can integrating deep

learning models (BERT, GPT-3), knowledge graphs, and user feedback mechanisms improve the effectiveness and personalization of AI-driven writing assessment systems, and how does this approach contribute to advancing current practices in automated writing evaluation? By investigating this question, the study contributes to the field by proposing a novel framework that enhances AI writing assessments through real-time, iterative feedback, making the system more adaptive to user behavior and context. This approach offers a more comprehensive solution to the limitations of existing writing evaluation systems.

## RELATED WORKS

A knowledge graph is a kind of information representation that uses nodes and edges to depict the connections between items in a domain and general knowledge about those entities. Wikidata, YAGO, Freebase, and DBpedia are all examples of large-scale knowledge graphs (*Vrandečić & Krötzsch, 2014*; *Suchanek, Kasneci & Weikum, 2007*; *Bollacker et al., 2008*; *Auer et al., 2007*). Knowledge graph embeddings are methods for extracting numerical representations of items and interactions from a knowledge graph into a feature space, where semantic linkages and patterns may be captured. There are several different ways to go about modeling. ConceptNet (*Speer, Chin & Havasi, 2017*) shows that one classical mathematical approach is to depict the relational space as a term-term matrix with positive pointwise mutual information and singular value decomposition. For instance, as seen in TransE (*Bordes et al., 2013*), another method relies on connection routes, which describe relations linearly as translations. Models such as GraphSAGE (*Hamilton, Ying & Leskovec, 2017*) and R-GCN demonstrate the alternative use of graph neural networks, which collect input from nearby entities using graph convolutional layers or comparable designs.

According to *Narciss et al. (2014)*, personalized tutoring feedback has the most educational relevance among the areas connected to computer-based technologies. To fix problems and monitor students' development, these writers looked at an intelligent learning environment that was accessible online and employed for mathematics. They concluded that there are gender-related variations in feedback efficiency for a few reasons. Firstly, females benefit more from tutoring feedback settings, particularly when the feedback is based on conceptual clues. Thirdly, males showed a rise in intrinsic drive, whereas girls increased their perceived competence faster.

*Jani et al. (2020)* shows how AI may be helpful for formative evaluation, assessment, and feedback using checklists and machine learning. According to its findings, using automated replies is a great way to monitor students' development and find ways to enhance clinical procedures. Because it uses AI to facilitate group projects, the *Santos & Boticario (2014)* experience stands apart. To encourage participation, debate, and teamwork as methods of instruction, they suggest an AI-based collaborative logical framework (CLF). In addition to making instructors' lives easier, they utilize this innovative assistance to keep tabs on how their kids are acting in class. Built specifically for the dotLRN e-learning platform, this adaptive guiding system aids in administrating and controlling student cooperation.

When it comes to feedback, there are some fascinating AI applications. Since it is the most basic explanation of how machines and people communicate, the idea of feedback has deep roots in cybernetics, control theory, and system theory; it also forms the basis of artificial intelligence. Using artificial intelligence, *Mirchi et al. (2020)* developed a Virtual Operative Assistant to provide trainees with automated feedback based on performance criteria during simulation-based medical training. They use VR and AI from a formative learning perspective to sort students according to competency performance standards, and then the system provides feedback to help them improve. The findings of *Janpla & Piriyasurawong (2020)* are also located in the same medical area. They create intelligent algorithms to choose questions for online examinations and study the application of AI to develop tests in e-learning settings. Data management, digital resource utilization or management, experiences with special education needs (SEN), intelligence system question answering, and other education-related activities are also available (*Gao et al., 2019*).

*Chatterjee & Bhattacharjee (2020)* examine AI to better manage their educational organizations' resources. *Liu et al. (2017)* use these kinds of technologies to assess the quality of schooling. Some writers (*Villegas-Ch, Arias-Navarrete & Palacios-Pacheco, 2020*; *Xiao & Yi, 2021*; *Castrillón, Sarache & Ruiz-Herrera, 2020*) suggest using AI to tailor course materials to individual students. Intelligent technologies for predicting academic performance are within our reach (*Troussas et al., 2023*). Improving individualised instruction is another area of interest for knowledge graphs. For example, *Troussas et al. (2023)* discovered that their knowledge graph-based tutoring system greatly enhanced students' learning effectiveness and productivity by suggesting tailored learning activities. By considering the crucial connections among knowledge points, *Lv et al. (2021)* developed a method based on weighted knowledge graphs to provide customized workout suggestions.

*Godwin-Jones (2022)* advocated adopting AWE tools like Grammarly, which use artificial intelligence to provide real-time feedback. These systems enhance the standards of writing by aiding users through the provision of predictive text and automated suggestions in the refinement process (*Kammer et al., 2023*). Described the use of artificial intelligence technologies in scientific writing, observing that AI tools assist in checking the accuracy and quality of scientific documents. Such AI-powered aids to writing assist researchers by refining the quality and meticulousness of their outputs (*Pasaribu, Budiman & Irawati, 2024*) reported the creation of a 1D CNN model to automate essay grading which achieved 81.18% accuracy in essay grading. This marks a progressive shift towards less subjectivity and burdening of educators while enhancing effortless grading accuracy and work efficiency.

*Bal & Öztürk (2025)* provided evidence for applying artificial intelligence technologies in writing skills enhancement and its effectiveness across all grades. Their study indicated that secondary school students benefited greatly from AI-supported systems that promote clear reasoning and writing improvement alongside critical thinking, thereby aiding K-12 students. Knowledge graphs (KGs) enable the complex relationships to be visualised, facilitating libraries, digital humanities, and writing concepts to be grasped more

effortlessly. The subsequent subsections explain the ways KGs are applied in the domain. KGs capture a network of communities which augment resource discovery and retrieval as emphasized by *Haslhofer, Isaac & Simon (2018)*. These community driven editorial processes enable KGs to be structured which enhances resource retrieval. Writing concepts and assessments can be improved by leveraging context-enhancing, community-driven editorial processes that enable KGs to be structured and give feedback. Writing concepts and assessments are made easier to contextually enhance feedback through parsing global KGs, which interconnect the world wide web. *Cheng (2020)* argued that the previously hidden intricate relationships are made explicit by KGs enabling effortless relationship searching, making comprehension easier when retrieving disparate resources. The processes can be crucial in generating automated tailored feedback for sophisticated writing assessments.

This process improves the model's learning of users' preferences by applying a global view of their interactions across the entire network (*Zuo et al., 2024*; *Wang et al., 2017*). Label imbalance is a frequent concern in massive datasets, where specific labels are disproportionately more common than others (*Ren et al., 2025*; *Li & Xing, 2025*). A graph is a mathematical construct comprising nodes (or vertices), units, and edges, which are connections between units (*Chen et al., 2024*; *Zhao et al., 2024*). These could include preprocessing data or performing dimensionality reduction to extract the most relevant features (*Li et al., 2025*; *Xu et al., 2025*). The result is commonly presented as a classification determining whether the graph or node is in-distribution (*Cao et al., 2025*; *Wang et al., 2023*). This makes word segmentation important for capturing the meaning of the text and more challenging for models to manage (*Chen et al., 2025*; *Shi, Dao & Cai, 2025*). A machine learning method where only a limited set of labeled data is used with a more extensive set of unlabeled data for training (*Liu et al., 2025*; *Song et al., 2025*). Those mechanisms guarantee that the motion adherence respects details like timing, gesture precision, and contextual features (*Wang et al., 2025*; *Li et al., 2023*). Clustering, classification, and regression techniques can be employed to analyse and predict spatiotemporal trends in the data (*Li et al., 2021*; *Ding et al., 2023*).

Integrating BERT and GPT-3 models for writing assessment is a well-established approach. Our study builds on this foundation by incorporating knowledge graphs and GNNs, which provide a deeper understanding of the relationships between writing concepts and offer more personalised feedback than traditional methods.

## METHODS AND MATERIALS

The AI writing assessment system is established starting from the first step, data collection, where writing samples and user feedback are gathered. After this, the pre-processing phase organizes the data and prepares it for analysis by cleaning it appropriately. As soon as the information is available, model selection is undertaken to identify a good algorithm for this application, including BERT and the GNN.

The overall processing flow is shown in Fig. 1. The training phase uses these models, which will help validate the data by learning to judge how well the writing is and provide

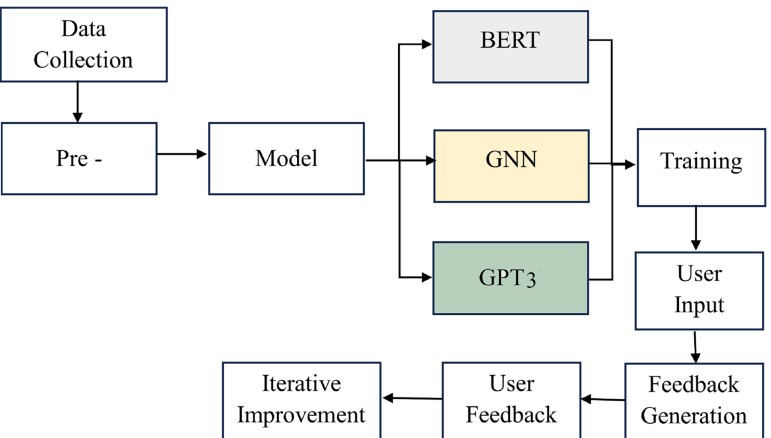

**Figure 1** The overall processing flow of the proposed methodology.

feedback on it. As expected, incorporating the GPT-3 model improves the system's efficiency and effectiveness in giving recommendations peculiar to the input components and the user's situational context. All this occurs through a cycle of ongoing increments, where users bring about effective changes to the system by providing feedback that makes corrections to the model and how feedback is given. Finally, this systematic and coordinated progression guarantees that the framework to conduct the AI writing assessment is efficient and assists users in improving their writing at higher levels.

## Data collection and preprocessing

The dataset used for training and testing the models comprised various writing samples, including published research papers, blog posts, and business documents. As illustrated in the pie chart, research papers constituted 40% of the dataset, while blog posts and business documents each represented 30%. This varied composition ensures that the models are exposed to different writing styles and contexts, enhancing their robustness and adaptability. Research articles were chosen primarily for their complexity and formal tone, while blog posts were chosen for their informal tone and complex vocabulary. Business documents were chosen for their professional style of writing. All samples were taken from datasets available publicly and were anonymised to uphold the privacy of the authors.

The dataset, $test_a ugmented.json$, comprises a structured collection of JSON objects, where each entry represents a unique query-answer pair and includes the following key fields:

Image Name ($imgname$): The file name of an image associated with the query, such as "$multi_col_803.png$". This association suggests that the image may contain relevant visual information that can aid in answering the query. Query (query): A natural language question about the content, such as "How many stores did Saint Laurent operate in Western Europe in 2020?". Label (label): The answer corresponding to the query, provided in a structured format. For instance, for the example query, the answer would be "47".

The imgname field indicates the specific image tied to each query, highlighting the potential role of visual data in providing or supporting the answer to the posed question. This association between the query and image allows for an exploration of how visual content may assist in accurate and contextually relevant responses.

The dataset used in this project is available on Kaggle and can be accessed *via* the following link, which includes a collection of tweets related to generative AI: https://www. kaggle.com/code/sanjushasuresh/generative-ai-creating-machines-more-human-like? select=GenerativeAI+tweets.csv

Additionally, user feedback was collected from a cohort of 100 participants who evaluated the usefulness and relevance of the AI system's suggestions. The bar chart reflects this feedback, showing that 50 participants found the suggestions "Very Useful," while 30 rated them as "Somewhat Useful." Only 20 participants considered the suggestions "Not Useful.". The selection criteria were based on past studies within the automated writing assessment branch ensuring diversity and statistical power to draw meaningful conclusions in the evaluation. Such a sample was also optimal for gathering comprehensive insights on user opinions concerning the AI suggestions. The 100 respondents who participated in the survey evaluating the AI writing assessment system were selected to ensure diversity in terms of language proficiency, age, education level, and experience with AI-based tools. Respondents had varying levels of English proficiency, including beginner, intermediate, and advanced learners. This diversity allowed us to assess the system's effectiveness across a broad range of language skill levels. The respondents ranged in age from 18 to 45 years, representing different stages of educational and professional backgrounds. Participants were from a wide array of educational disciplines, including students (both undergraduate and graduate), professionals, and ESL teachers. This variety ensured that the feedback represented multiple perspectives on the AI system's usefulness. Of the 100 respondents, approximately 30% had prior experience using AI-based writing tools (such as Grammarly), while the remaining 70% were new to such technologies. This distinction allowed us to evaluate the system's effectiveness for novice and experienced users. Figure 2 shows the data distribution of data collection.

These results highlight the effectiveness of the AI writing assessment framework in providing valuable feedback, contributing to a positive user learning experience, and emphasizing the importance of continuous improvement in the system.

Every dataset comes with possible biases impacting a model's effectiveness, and in this case, we identified the following: Genre bias: This dataset may be overly dominated by some genres, for instance, academic writing, which may impair the model's genre evaluation capabilities. We tried to solve this problem by ensuring the dataset had balanced proportions of different styles (research papers, blog posts, and business documents). These data collection processes also included diverse sets from each genre to minimize potential genre bias. Cultural bias: Since most dataset samples were taken from documents written in English, the feedback's writing style and vocabulary face the risk of a culture bias. We tried to eliminate this bias by drawing from various documents from different regions and cultures, ensuring that the AI models would be exposed to various writing

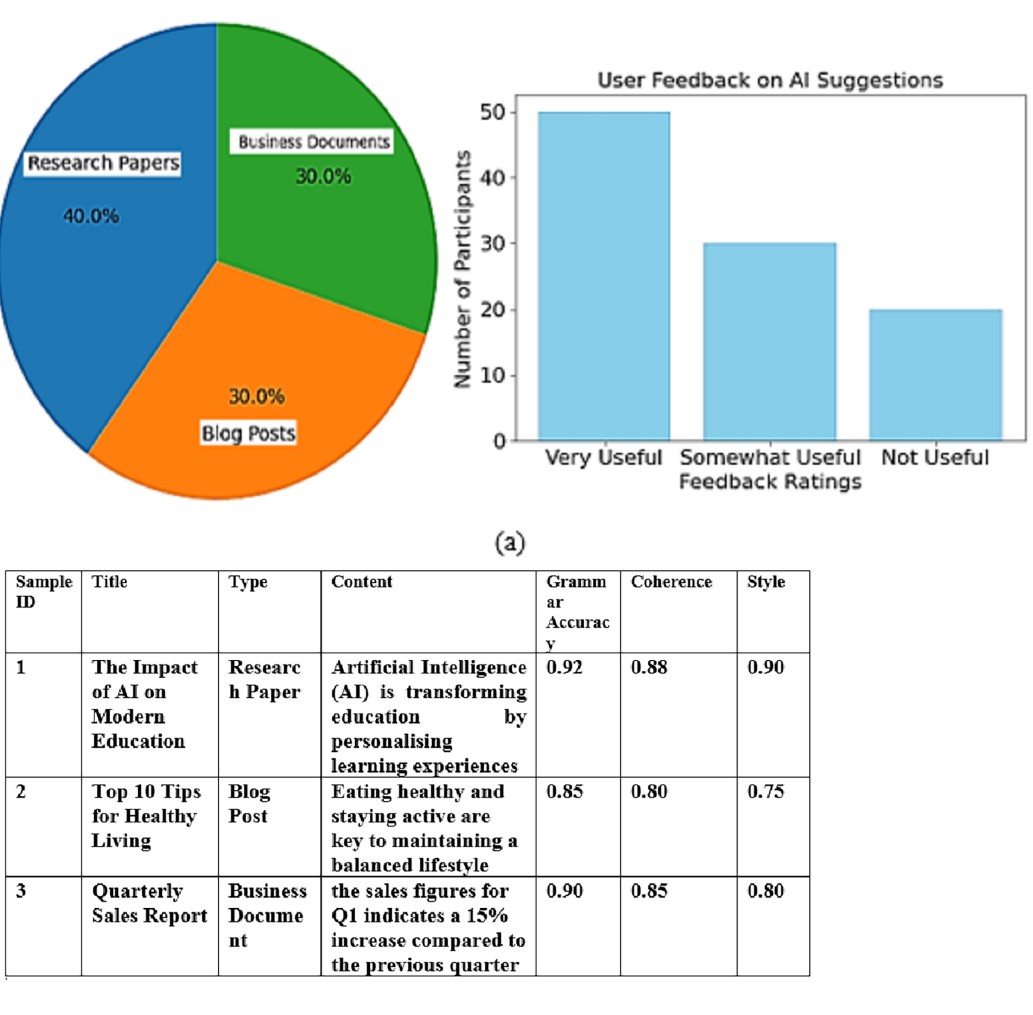

| Sample ID | Title | Type | Content | Grammar Accuracy | Coherence | Style |
|---|---|---|---|---|---|---|
| 1 | The Impact of AI on Modern Education | Research Paper | Artificial Intelligence (AI) is transforming education by personalising learning experiences | 0.92 | 0.88 | 0.90 |
| 2 | Top 10 Tips for Healthy Living | Blog Post | Eating healthy and staying active are key to maintaining a balanced lifestyle | 0.85 | 0.80 | 0.75 |
| 3 | Quarterly Sales Report | Business Document | the sales figures for Q1 indicates a 15% increase compared to the previous quarter | 0.90 | 0.85 | 0.80 |

(b)

**Figure 2** (A) Data distribution of data collection (B) Sample collected data.

styles and vocabularies. Demographic bias: If the dataset leaned heavily towards the writings from certain demographic groups, such as age or level of education, the model might develop a bias toward those specific writing styles. For data collection, we obtained documents from various sources such as academic journals, blogs, and business websites, which ensured representation from a wide range of demographic groups. Additionally, we aimed to gather feedback from participants from different backgrounds to mitigate demographic bias in the feedback. As writing samples from the data set were created with manual intervention, there is an inherent risk of the writing features, such as grammar, coherence, and vocabulary, being affected due to subjective biases bound to personal preferences of a certain set of annotators. To mitigate the concerns of annotation bias, the process involved multiple annotators, and the annotations were checked for reproducing reliable results, such as grammar, coherence, and vocabulary, and empirically consistent results. Furthermore, participant feedback was scrutinised to expose potential gaps where

human judgment could introduce discrepancies, and such biases were systematically addressed. The primary aim was to reinforce the reliability of the data, which is why thorough validation processes were undertaken. The data set was partitioned into training (80%) and test (20%) sets to guarantee that the models would be assessed based on entirely independent data. In addition, the training phase included cross-validation methods, which further tested the model's claim of performing on multiple subsets of the data and offered a more substantial claim of evaluating the model's generalizability. In our assessment, implementing all measures aimed at addressing biases enhances the credibility of the dataset in this study for designing and assessing the efficacy of AI models. These approaches put the dataset through a variety of tests, which strengthens the primary hypothesis that regardless of the context or angle from which the data is approached, the adaptable nature of the model will allow for dependable outcomes.

## Integration of techniques and models

To integrate new advanced techniques, such as deep learning models, knowledge graphs, and graph neural networks (GNNs) into a cohesive system, this study tackles the problem of personalized feedback. It incorporates all of them working together to analyze the feedback addressed to an individual's document. All components have their unique contribution yet work together through the AI-powered writing assessment system. Writing samples are processed through deep learning models BERT and GPT-3, which provide insights considering the divergent factors like grammar, coherence, and vocabulary. Feedback in this case can be derived using knowledge graphs as they offer a structural representation of the concepts pertaining to a certain document, alongside their interrelations. Lastly, GNNs have been used to process the interpersonal data and relational data in the knowledge graph to the writing sample to enhance the construction of the complex dependencies of writings, in this case.

In this section, we explain and describe the individual model mechanisms and interactions with the data to enhance relevance and precision in determining the assessment. We also outline the other outlined models to show how the components were integrated. These models have been incorporated for additional tailoring of the feedback, which can shift according to the system requirements and unique needs of the ESL users.

## BERT model

*Devlin et al. (2019)* created BERT (Bidirectional Encoder Representations from Transformers), a crucial transformer model. This advancement has benefited the field of natural language processing, and models may examine not only the words that came before the current word but also their connection to every other word in the sentence. This feature of BERT and its training methods make it very effective for many linguistic tasks, including text categorisation, question-and-answer, and writing assessment.

Stacked encoder transformers constitute the basis of the BERT architecture. Every encoder in the BERT design incorporates a feedforward network, residual connections, layer normalisation, and a multi-head self-attention mechanism. Three types of

embedding—the token, the segment, and the position—make up this input representation in BERT. Figure 3 shows the BERT model architecture.

In token embeddings, a trained matrix transforms each word supplied into a vector. Segment embeddings aid comprehension for activities involving two sentences by dividing the two phrases into A and B. Due to the lack of a built-in method for encoding location, positional embedding transformations often exclude the positional embeddings that specify the token sequence.

The final input embedding for the token $i$ can be expressed as:

$$e_i = w_i + p_i + t_i \tag{1}$$

where, $w_i$ is the token embedding, $p_i$ is the positional embedding, $t_i$ is the segment embedding.

When encoding each token, the self-attention mechanism lets BERT weigh the significance of different words in the input sequence. The attention scores between tokens $i$ and $j$ are calculated as follows:

$$\alpha_{ij} = \frac{exp(Q_i \cdot K_j^T)}{\sum_{k=1}^{n} exp(Q_i \cdot K_k^T)} \tag{2}$$

where, $Q_i$ is the query vector for the token $i$. $K_j^T$ is the key vector for the token $j$. $n$ is the total number of tokens in the sequence. The output representation for each token is computed as a weighted sum of the value vectors:

$$z_i = \sum_{j=1}^{n} \alpha_{ij} V_j \tag{3}$$

where $V_j$ is the value vector associated with the token $j$. BERT is pre-trained on a large *corpus* of text using two primary tasks. In the masked language model (MLM) task, some tokens in the input sequence are randomly masked, and the model learns to predict these masked tokens based on the surrounding context. The loss function for MLM is defined as:

$$L_{MLM} = -\sum_{i \in M} logP(x_i | x_{context}) \tag{4}$$

where $M$ is the set of masked positions, $x_i$ are the masked tokens, and $x_{context}$ refers to the other tokens in the sequence. The next sentence prediction (NSP) task predicts whether a given sentence B follows sentence A in the original text. The loss for NSP is defined as:

$$L_{NSP} = -\sum_{i=1}^{N} (y_i logP(isnext) + (1 - y_i) \log P(notnext)) \tag{5}$$

$y_i$ indicates whether sentence B follows sentence A (1 for true, 0 for false). After pre-training, BERT can be fine-tuned for specific tasks such as writing assessments. A task-specific output layer (*e.g.*, softmax) is added to predict writing quality metrics. The

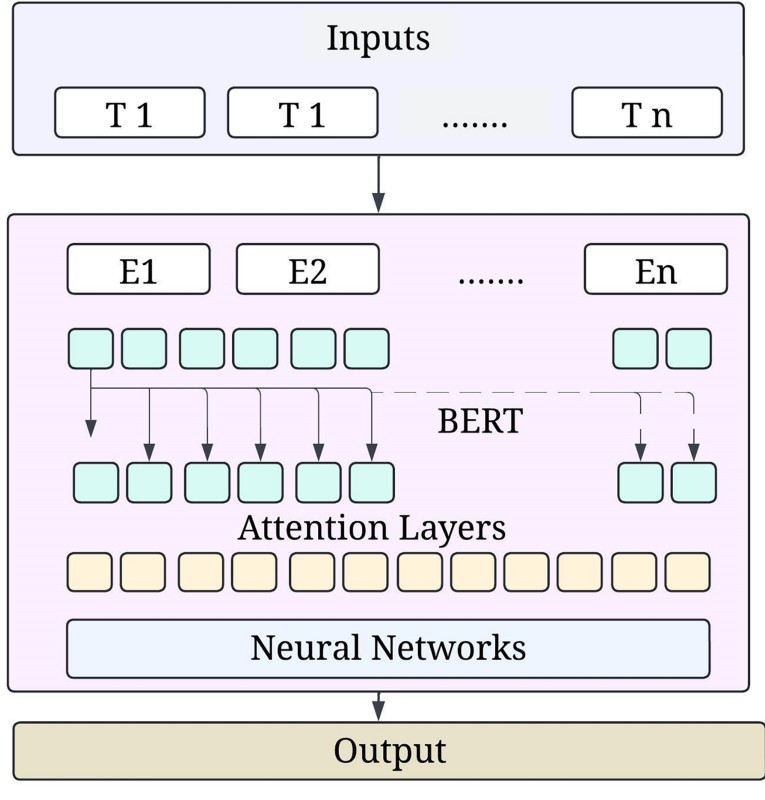

**Figure 3 BERT model architecture.**

entire model, including the pre-trained BERT weights, is fine-tuned on the annotated writing samples using backpropagation to minimise the cross-entropy loss:

$$L = -\sum_{t=1}^{N} \log P(y_t|x) \tag{6}$$

where $y_t$ is the actual label for the writing sample, and $x$ is the input representation. BERT's architecture and training methodology allow it to excel in understanding context and semantics in writing. By leveraging its bidirectional encoding and attention mechanisms, our framework effectively evaluates writing quality, providing valuable insights and personalised feedback to users. Integrating BERT into the writing assessment system enhances the ability to capture the intricacies of language, ultimately contributing to improved writing skills for learners.

## GPT model

GPT-3, which stands for Generative Pretrained Transformer-3, is an advanced natural language model engineered by OpenAI. It is important to note that thanks to 175 billion parameters, it is clear that GPT-3 can easily comprehend context and structure and generate text similar to that of a human. Its approach to feeding data takes an

autoregressive format to predict successive data. Hence, it is helpful in text completion and feedback, dialogue systems and other areas requiring interactive writing. Built upon a sequence of decoder layers, GTP-3 uses the structure of the transformers. GPT-3 uses decoder stack architecture and focuses on text generation and completing sequences of unfilled text through the following token prediction mechanism instead of BERT, which builds on the encoder stacks. Every decoder layer has a self-attention mechanism and a feedforward neural network mechanism, followed by a layer normalisation and a residual connection.

The input to GPT-3 is a sequence of tokens converted into embeddings. Each token embedding $e_i$ is represented as follows:

$$e_i = w_i + p_i \tag{7}$$

where, $w_i$ is the token embedding. $p_i$ the positional embedding helps the model understand the order of tokens in the sequence. The self-attention mechanism in GPT-3 enables the model to focus on relevant parts of the input sequence when generating the next token. The attention score $\alpha_{ij}$ between tokens $i$ and $j$ is computed similarly to BERT:

$$\alpha_{ij} = \frac{exp(Q_i \cdot K_j^T)}{\sum_{k=1}^{n} exp(Q_i \cdot K_k^T)} \tag{8}$$

where, $Q_i$ is the query vector for the token $i$. $K_j^T$ is the key vector for the token $j$. $n$ is the total number of tokens in the sequence. The output representation for each token is computed as a weighted sum of the value vectors:

$$z_i = \sum_{j=1}^{n} \alpha_{ij} V_j \tag{9}$$

where $V_j$ is the value vector associated with the token $j$. GPT-3 operates autoregressively, generating text one token at a time based on the previously generated tokens. The probability of developing a sequence of tokens $x = (x_1, x_2, \ldots, x_n)$ is defined as:

$$P(x) = \prod_{t=1}^{n} P(x_t | x_{<t}) \tag{10}$$

where $x_{<t}$ denotes all tokens preceding $x_t$. The output for each token is generated using the softmax function over the log produced by the model:

$$P(x_t | x_{<t}) = \frac{exp(y_t)}{\sum_{v \in V} exp(y_v)} \tag{11}$$

where $y_t$ represents the output logits for the token $t$, and $V$ is the vocabulary size.

The model is trained to minimise the cross-entropy loss between the predicted token probabilities and the actual tokens in the training data. The loss $L$ can be expressed as:

$$L = -\sum_{t=1}^{n} \log P(x_t | x_{<t}) \tag{12}$$

This loss function quantifies how well the model predicts the next token in the sequence.

GPT-3 can be fine-tuned for specific applications, such as writing assessments, by adapting its training objective to include user feedback and context. Task-Specific Prompting provides task-specific prompts that guide GPT-3 in generating relevant feedback and suggestions based on the input writing sample. The reinforcement learning from human feedback (RLHF) approach incorporates human feedback into the training process to improve the model's performance on specific tasks. The objective can be adjusted to maximise user satisfaction and writing quality.

Although GPT-3 is generally not fine-tuned post-release, using prompts can effectively leverage its capabilities for specific tasks. However, in a research context, if the model were to be fine-tuned, the weight updates would follow:

$$\theta \leftarrow \theta - \alpha \nabla L(\theta) \tag{13}$$

where $\theta$ represents the model parameters, $\alpha$ is the learning rate and $\nabla L(\theta)$ is the gradient of the loss for the parameters.

Due to GP-3's autoregressive architecture and the extensive pre-training it has undergone, it is good at tasks involving the generation of contextually appropriate text. Understanding how the AI works, it is possible to achieve customised feedback and promote user interaction when incorporating GPT-3 into the AI writing assessment system and ensuring the progress of writing skills. Flexible understanding of writing context makes GP-3 potent for enhancing writing teaching and assessment.

## Graph neural networks

GNNs are a family of neural networks constructed to treat graph-structured data as the primary input. Convoluted network models and other traditional models deal only with grid-like information, such as images or sequences. However, GNNs fully exploit the relations encoded in graphs. Therefore, they are naturally suited for representing and working with inherently relational data. Some examples include social networks, molecular structures, and knowledge graphs. GNNs model, as shown in Fig. 4, uses both the node and edge properties to represent how different entities are interconnected in the graph. In this scenario, GNNs will be used to improve the graphic integration of knowledge in this writing quality assessment AI between concepts so that the model can use these concepts in writing and about one another.

A graph $G$ is represented as a pair $(V, E)$, where, $V$ is a set of nodes (or vertices) representing entities or concepts (*e.g.*, writing techniques, themes). $E$ is a set of edges representing relationships between the nodes (*e.g.*, "supports," "enhances," "contributes to"). The graph can also be represented with an adjacency matrix. $A$, where each element $A_{ij}$ indicates the presence or absence of an edge between nodes $i$ and $j$. GNNs operate through message-passing and aggregation steps, allowing nodes to update their representations based on their neighbours' information.

GNNs operate through message-passing and aggregation steps, allowing nodes to update their representations based on their neighbours' information. During each iteration

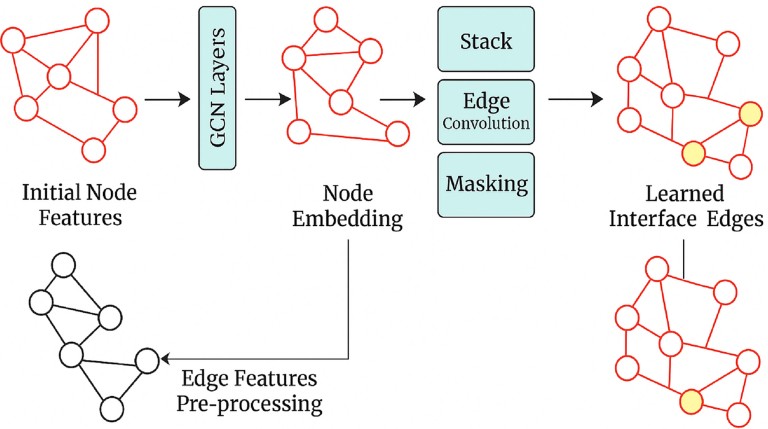

**Figure 4  GNN model architecture.**

$t$, each node $v_i$ in the graph aggregates messages from its neighbours. The message for the node $i$ is computed as:

$$m_i^{(t)} = \sum_{j \in N(i)} W^{(t)} \cdot h_j^{(t-1)} \tag{14}$$

where, $m_i^{(t)}$ is the aggregated message for node iii at iteration $t$. $N(i)$ is the set of neighboring nodes connected to the node $i$. $W^{(t)}$ is the weight matrix for the current layer. $h_j^{(t-1)}$ is the feature representation of the neighbour $j$ from the previous iteration. After message passing, each node updates its representation based on the aggregated messages and its previous state,

$$h_i^{(t)} = \sigma\left( W^{(t)} \cdot m_i^{(t)} + b^{(t)} + h_i^{(t-1)} \right) \tag{15}$$

where, $h_i^{(t)}$ is the updated feature representation for the node $i$ at iteration $t$, $\sigma$ is a non-linear activation function (*e.g.*, ReLU), $b^{(t)}$ is a biased term. A popular variant of GNNs is graph convolutional networks (GCNs), which generalise the concept of convolution from Euclidean spaces to graph-structured data. The layer-wise propagation rule for GCNs can be expressed as:

$$H^{(l+1)} = \sigma\left( \tilde{A} H^{(l)} W^{(l)} \right) \tag{16}$$

where, $H^{(l)}$ is the feature matrix of the graph at the layer $l$, $\tilde{A} = D^{-\frac{1}{2}} A D^{-\frac{1}{2}}$ is the normalised adjacency matrix with self-loops added, and $D$ is the degree matrix, $W^{(l)}$ is the weight matrix for the layer $l$, $\sigma$ is the activation function. GNNs enhance the model's understanding of relationships within knowledge graphs in our AI writing assessment framework. By learning representations that capture the interdependencies between writing concepts, GNNs facilitate context-aware feedback generation.

  For example, if a user writes about "effective communication," the GNN can identify related concepts (*e.g.*, "clarity," "engagement") within the knowledge graph and enable the AI system to provide targeted suggestions. This relationship learning enhances the

relevance and quality of the user feedback. The training of GNNs typically involves a supervised learning approach where the model learns to predict labels or outputs associated with nodes in the graph. The loss function can be defined as:

$$L = -\sum_i y_i \log\left(P\left(y_i|h_i^{(L)}\right)\right) \tag{17}$$

where, $y_i$ is the actual label for the node $i$, $\left(P(y_i|h_i^{(L)})\right)$ is the predicted probability of the label given the node's representation at the final layer L. Graph neural networks are instrumental in enriching our AI writing assessment architecture with structured relational data. The learning from relationships represented in knowledge graphs enhances the model's competence in generating contextually appropriate feedback. This addition improves the understanding of writing quality and improves writing by providing relevant feedback on the surface and deeper aspects of the writing.

## HYPERPARAMETER TUNING

Additional factors to be optimized for the AI writing assessment framework's hyperparameter application include the GPT-3 model, BERT, and GNNs (Table 1). Starting with BERT, the learning rate is 2e−5, which guarantees fast enough training and partial convergence. Additionally, for best training efficiency, the batch size should be 32. For sufficient data consumption, training is conducted over around 50 epochs, and to accommodate lengthier writing inputs, the maximum sequence length is set at 256 tokens. In addition, a 10% dropout is employed to limit overfitting.

In order to facilitate steady training, a learning rate of 0.001 has been chosen for GNNs. To quantify the intricate interplay present in the data, three tiers are employed. The model's optimal hidden unit size for learning capacity was 128. A non-linear activation function that can include non-linearity into the model is the ReLu function. A 20% dropout is also a part of it to make it more regular.

Table 1 presents further considerations for applying hyperparameters in the AI writing assessment framework concerning BERT, GNNs, and the GPT-3 model, which would require optimisation. Starting with BERT, the learning rate is defined by 2e−5, which ensures sufficiently fast training and some convergence. Furthermore, the batch size is optimal at 32 for increased training efficiency. The training is done for about 50 epochs to ensure enough data consumption, and the maximum sequence length is set to 256 tokens to allow for longer writing inputs. Furthermore, a dropout of 10% is also used to restrict overfitting.

Regarding GNNs, a learning rate of 0.001 has been adopted, which enables stable training. Three levels are used to measure complicated interactions embedded within the data. A hidden unit size 128 was the optimum for the model's learning capacity. Similarly, the ReLu function is chosen as a non-linear activation function because it incorporates non-linearity into the model. Also included is a 20% dropout to increase the regularisation.

To achieve rapid convergence for the GPT-3 model, a learning rate 5e−5 and a batch size 16 are implemented to ensure sufficient input data representation. The model is trained for 50 epochs, which is reasonable for a fine-tuned model. The maximum sequence

**Table 1 Hyperparameter tuning table.**

| Model | Hyperparameter | Optional values | Final values |
|---|---|---|---|
| **BERT** | Learning rate | 1e−5, 2e−5, 3e−5, 5e−5 | 2e−5 |
| | Batch size | 16, 32, 64 | 32 |
| | Epochs | 3, 5, 10 | 5 |
| | Max sequence length | 128, 256, 512 | 256 |
| | Dropout rate | 0.1, 0.2, 0.3 | 0.1 |
| **GNN** | Learning rate | 0.01, 0.001, 0.0001 | 0.001 |
| | Number of layers | 2, 3, 4 | 3 |
| | Hidden units | 64, 128, 256 | 128 |
| | Activation function | ReLU, Sigmoid, Tanh | ReLU |
| | Dropout rate | 0.1, 0.2, 0.3 | 0.2 |
| **GPT-3** | Learning rate | 5e−5, 1e−4, 1e−5 | 5e−5 |
| | Batch size | 8, 16, 32 | 16 |
| | Epochs | 3, 5, 10 | 3 |
| | Sequence length | 128, 256, 512 | 256 |
| | Temperature | 0.7, 0.9, 1.0 | 0.9 |
| | Top-k sampling | 5, 10, 20 | 10 |
| | Top-p (nucleus) sampling | 0.9, 0.95 | 0.95 |

length is 256 tokens to handle long texts effectively. To ensure that the generated outputs exhibit good coherence and relevance, a temperature of 0.9, a top-k value of 10, and a top-p (nucleus) value of 0.95 are used. These carefully selected hyperparameters aim to enhance the model's ability to provide relevant and precise writing feedback.

We train the GPT-3 model for 50 epochs, which is reasonable for a fine-tuned model, to achieve rapid convergence. We set the maximum sequence length to 256 tokens to handle long texts effectively. To make sure the generated outputs are coherent and relevant, we use a temperature of 0.9, a top-k value of 10, and a top-p (nucleus) value of 0.95. These hyperparameters ensure the model can provide relevant and precise writing feedback.

## RESULT AND DISCUSSION

The experiment was configured on a machine that had these specifications: the operating system was Ubuntu 20.04 LTS, an Intel i7-9700K CPU with eight cores at 3.6 GHz, 32 GB of DDR4 RAM, an NVIDIA RTX 2080 Ti with 11 GB of VRAM, and a 1 TB SSD. In addition to the 11 GB of VRAM provided by the NVIDIA graphics card, the system had 32 GB of DDR4 RAM, which allowed for quick storage of temporary data. The system also had a 1 TB SSD, enabling fast access to stored data. The software environment included deep learning frameworks such as TensorFlow 2.5 and PyTorch 1.9. Specific model libraries also included Hugging Face Transformers for BERT and GPT-3 and the deep graph library (DGL) for GNNs. The version of Python that was installed was 3.8.10 and the system had CUDA 11.2, which allowed for GPU acceleration. As well as the model training parameters, every model was evaluated based on four fundamental metrics: accuracy,

**Table 2  BERT model performance metrics.**

| Metric | Value (%) |
| --- | --- |
| Accuracy | 96.5 |
| Precision | 95.8 |
| Recall | 96.2 |
| F1-score | 96.0 |

**Table 3  GNN model performance metrics.**

| Metric | Value (%) |
| --- | --- |
| Accuracy | 97.2 |
| Precision | 96.5 |
| Recall | 97.0 |
| F1-score | 96.7 |

precision, recall, and F1 score. These metrics were chosen to ensure each model was assessed for relevance, accuracy, and usefulness to, and the system had CUDA 11.2, the user feedback effectiveness. Accuracy measures the total correctness of model feedback. Precision evaluates the level of false positive avoidance achieved by the model. Recall assesses the model's coverage for all relevant writing issues. The F1 score evaluates the precision and recall in a harmonic mean, balancing both.

Each model's result is presented below in the same order, along with the performance results metric in the appropriate table for each model. As for the BERT model's accuracy level, it was very high at 96.5%, which was outstanding. This shows how well it can detect grammatical errors and provide helpful feedback. A high value with a precision of 95.8% is justified as the number of false positives suffered was low, yielding a high number of positive cases instead. Apart from that, with a recall of 96.2%, neither false negative was experienced as most of the relevant instances were covered. Thus, a good F1 score of 96.0 was attained, demonstrating a good balance between their precision and recall ability.

Table 2 summarizes the performance metrics for the BERT model.

The third model, the GNN model, recorded the highest accuracy out of the three at 97.2%. Such excellent performance speaks to the active performance of the model in properly leveraging connections and relationships presented within the knowledge graph. Similarly, the results of the GNN with a precision of 96.5% and recall of 97.0% stood tall on all these metrics, resulting in a strong F score of 96.7. This means that the GNN places writing feedback in context, and users can expect helpful feedback regarding writing. Table 3 summarises the performance metrics for the GNN model.

Despite being appreciably lower at a rate of 95.6% overall accuracy of the GPT model, it did well enough regardless. It is competent in providing coherent and contextually relevant suggestions aligned with the user-made ones. The precision score of 95.2% means that the model exhibits a relatively low false alarm rate, and the recall score of 95.5% shows the model's efficiency in recognition of all the relevant instances of writing. The resultant F1

**Table 4 GPT-3 model performance metrics.**

| Metric | Value (%) |
|---|---|
| Accuracy | 95.6 |
| Precision | 95.2 |
| Recall | 95.5 |
| F1-score | 95.4 |

score of 95.4% indicated no severe dereliction of duty towards overload or underload. The model performed well in the writing quality feedback mechanism. Table 4 summarizes the performance metrics for the GPT-3 model.

The comparative analysis of the three models indicates that all performed very well and all recorded accuracy rates of more than 95%. The lead was occupied by the GNN model, where an accuracy of 97.2% was achieved, showcasing the ability to grasp relationships in the knowledge graph. The BERT model stood second with 96.5% accuracy, pinpointing grammatical and contextual mistakes. On the other hand, the performance of the GPT-3 model was slightly less, getting 95.6%, but it contributed well to coming up with relevant and appropriate suggestions. Figure 5 shows the accuracy achieved by various models.

Table 5 and Fig. 6 summarise the findings from the error analysis, feature importance analysis, and time efficiency metrics for the AI writing assessment framework. The error analysis suggests that all models had many writing problems, with the recurring problem being grammar. It was also observed that the BERT model made an error to the extent of 60% of the grammatical errors, mainly the subject's agreement with the verb and the formulation of tenses the other way around.

The GNN model equally had a fair number of grammatical errors, 55%, typical of which was probably incorrect punctuation. The figure was slightly better for GTP-3, as only half of the output presented errors of the grammatical type with incorrect word choice, but this still constituted 50%. There were noticeable coherence errors in GTP-3 and output that concerned improving the language structure through the better utilisation of paragraphs and breakages in long sentences.

Feature importance analysis from Table 6 appears to have clarified that grammar is the most emphasised, with 40% of the total importance score. A total of 30% for coherence followed in order of importance, which underscored the importance of this feature in providing feedback on writing quality. Also, it was relatively essential but not as much; in rank order, vocabulary richness was 15% and style 10%.

The overall clarity of writing was undoubtedly the least contributing feature, with only 5%. This understanding implies that grammatical use and coherence, especially in the Boston Bulger speech used for persuasive purposes, would be the norm in polite society's writing. Yet, opportunities still exist to elevate what was ordered to be stylistic and vocabulary richness in the ensuing models' versions.

In the feature importance analysis presented in Fig. 7, particular focus is placed on various aspects of writing quality in the AI writing evaluation system and their
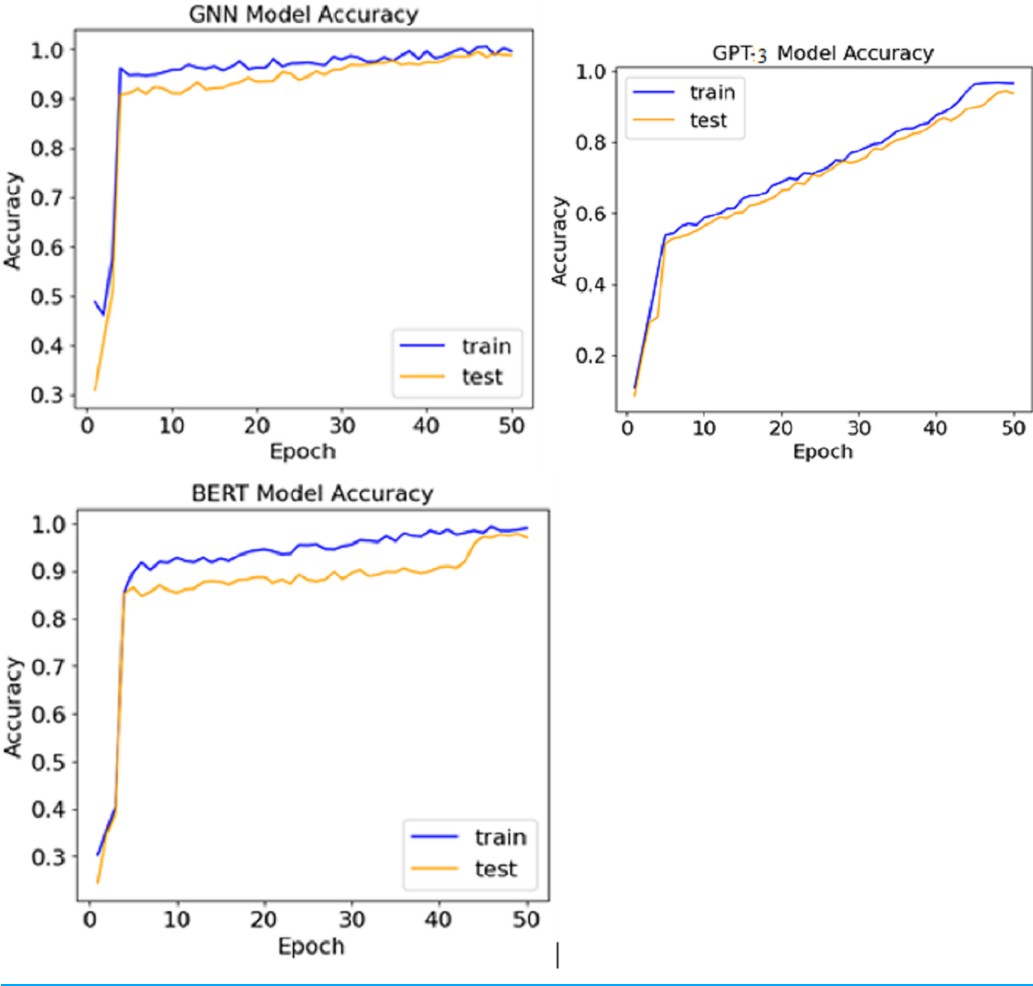

**Figure 5** Accuracy achieved by various models.

**Table 5 Error analysis.**

| Model | Type of error | Frequency (%) | Common examples |
|---|---|---|---|
| BERT | Grammatical errors | 60 | Subject-verb agreement, tense misuse |
| | Coherence issues | 25 | Lack of logical flow |
| | Style inconsistencies | 15 | Inconsistent tone |
| GNN | Grammatical errors | 55 | Punctuation errors |
| | Coherence issues | 30 | Topic relevance issues |
| | Style inconsistencies | 15 | Repetitive phrasing |
| GPT-3 | Grammatical errors | 50 | Incorrect word choice |
| | Coherence issues | 35 | Disjointed sentences |
| | Style inconsistencies | 15 | Overly complex sentences |

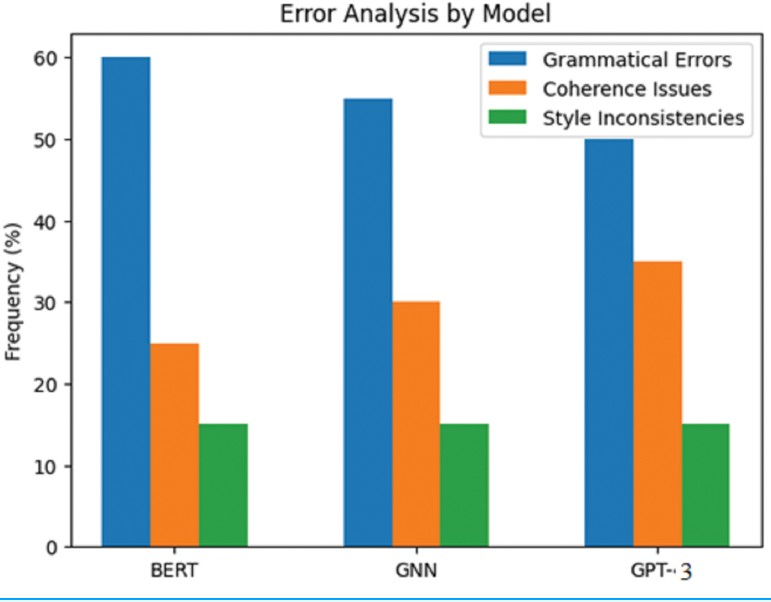

**Figure 6 Error analysis by model.**

**Table 6 Feature importance analysis.**

| Feature | Importance score (%) |
|---|---|
| Grammar | 40 |
| Coherence | 30 |
| Vocabulary richness | 15 |
| Style | 10 |
| Overall, Clarity | 5 |

significance. The picture is presented in a pie chart, and the reader can appreciate the features that contributed to the overall decision-making.

Figure 8 depicts the corresponding time efficiency metrics of the AI writing assessment framework in terms of average processing time, feedback generation time, and total time taken for BERT, GNN, and GPT-3 models. Each model's time efficiency in terms of performance has been clearly illustrated with a line graph.

Results within Fig. 8 show that, on average, each model takes some processing time; however, on average, BERT seems to take the least amount of time, with an average time of around 0.45 s. This suggests that BERT is swift in examining writing samples and giving appropriate feedback. The GNN model is not far behind, with an average processing time of 0.50 s, which is quite fair because it has to learn the relationships from the knowledge graph. The GPT-3 model takes approximately 1 min on average, much longer than the rest. This prolonged period is likely because the model's generative accessories cannot para-hybridize contextually rich and coherent responses without an increase in
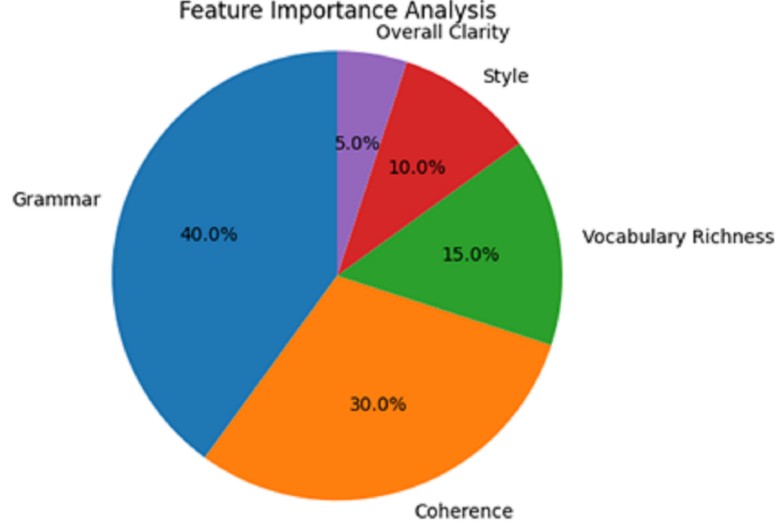

**Figure 7  Feature importance analysis.**     

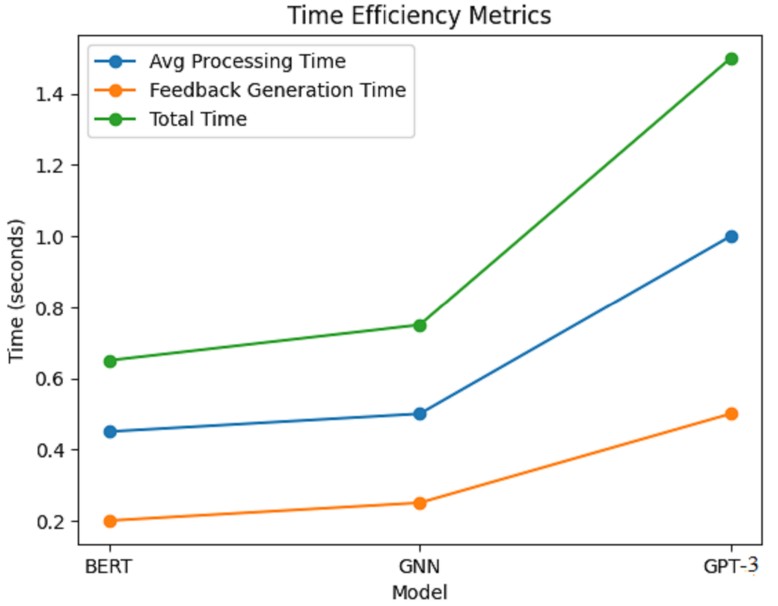

**Figure 8  Time efficiency metrics.**     

**Table 7  Time efficiency metrics.**

| Model | Average processing time (s) | Feedback generation time (s) | Total time (s) |
|---|---|---|---|
| BERT | 0.45 | 0.20 | 0.65 |
| GNN | 0.50 | 0.25 | 0.75 |
| GPT-3 | 1.00 | 0.50 | 1.50 |

**Table 8 Model performance comparison.**

| Model | Accuracy (%) | Precision (%) | Recall (%) | F1-score (%) |
|---|---|---|---|---|
| Rule-based grammar checker | 75.0 | 70.0 | 65.0 | 67.5 |
| Logistic regression | 80.0 | 75.0 | 70.0 | 72.5 |
| Support vector machine | 85.0 | 80.0 | 75.0 | 77.5 |
| BERT | 96.5 | 95.8 | 96.2 | 96.0 |
| GNN | 97.2 | 96.5 | 97.0 | 96.7 |
| GPT-3 | 95.6 | 95.2 | 95.5 | 95.4 |

**Table 9 Paired t-test results and effect size comparison.**

| Model comparison | $p$-value (%) | Cohen's d (Effect size) (%) |
|---|---|---|
| BERT *vs.* GPT-3 | 0.03 | 0.58 (Medium) |
| BERT *vs.* GNN | 0.01 | 1.22 (Large) |
| GPT-3 *vs.* GNN | 0.45 | 0.12 (Small) |

computational effort. For the GNN model, it was a little slower, averaging 0.75 s, due to the complexity of its relational learning mechanisms, as shown in Table 7.

Considering feedback generation time, again, BERT delivers the feedback in the shortest time out of the three models; on average, this recovery took about 0.20 s. GNN spent about 0.25 s generating feedback, while it took the longest, with GPT-3 at an average of 0.50 s. The total time is a statistic that accounts for the average processing time, feedback generation time, and feedback generation time: BERT with 0.65 s, GNN with 0.75 s, and GPT-3 who took 1.50 s. Regarding time efficiency, BERT was the most time-efficient model, averaging 0.65 s to generate feedback. Thus, the advantages of the new models over the baseline approaches are substantial compared to the baseline approaches, as shown in Table 8.

The paired t-test outcomes shed light on the differences in the model performance metrics more quantitatively, as shown in Table 9. The Rule-Based Grammar Checker was the tool that performed the worst, with the lowest accuracy of just 75%. This limitation indicates the problems associated with using such an approach, such as the imposition of many rules on a language, which is usually quite complex and intricate. The logistic regression model managed to improve the results concerning the models as it attained a level of 80% in terms of accuracy rating, while the support vector machine's rate was 85%. However, both traditional models still come short as advanced models are concerned, particularly concerning precision and recall. The accuracy of 96.5% achieved by the BERT model coincided with high precision and recall scores. The outstanding performance translates to BERT having a good comprehension and relating appropriate responses to the situations. The model based on the GNN approach surpassed the BERT, reaching an accuracy of 97.2%, confirming the effectiveness in exploiting the relational data. The low accuracy of 95.6% in the case of GPT-3 compared to BERT did not change the case as there

were significant outcomes, particularly in the ability to provide contextual and logical suggestions.

## Statistical analysis

We executed paired t-tests to evaluate the performance of the models including BERT, GPT-3, and GNN on various metrics. Several intriguing insights were provided from the data. During the evaluation of the performance metrics of BERT and GPT-3, the difference observed was accuracy and it was significant $t(12) = 2.389$, $p = 0.03$ with BERT having better accuracy than GPT-3. Also, in evaluating BERT and GNN, the accuracy difference was statistically significant, with GNN outperforming BERT at $p = 0.01$. However, the GPT-3 and GNN comparison yielded a $p$-value of 0.45, which indicates no significant difference. Furthermore, we computed Cohen's d to evaluate the impact of the biological or clinical significance of these differences. The effect was significant for the difference between BERT and GNN and moderate for the difference between BERT and GPT-3.

For instance, the assessment of BERT against GPT-3 showed a significant difference of 3%, with GPT-3 having the lower score. This means that BERT proved more proficient than GPT-3 in evaluating the writing quality during the experiment. Likewise, BERT was compared to GNN and showed a significant accuracy gain for GNN with lower accuracy in BERN having a $p$-value of 0.01, which makes GNN the dominant model. This is a good illustration of the ability of Graph Neural Networks to capture the complicated relations in writing data to improve system feedback.

In contrast, the assessment of GPT-3 against GNN produced a $p$-value of 0.45, indicating that the models did not differ significantly in performance. This means that, although GNN performed better than GPT-3 in some respects, the differences were not robust enough to be deemed statistically significant. To evaluate the magnitude of differences, Cohen's d values were computed. The difference in effectiveness between BERT and GNN was large (1.22), indicating BERT was significantly outperformed by GNN. While the difference in effectiveness between BERT and GPT-3 was moderate (0.58), there was still a meaningful difference in accuracy. However, the small effect size (0.12) between GPT-3 and GNN strengthens the conclusion that the performance difference was not substantial. These statistical measures, when taken together, provide greater insight into the performance of the models while AI-based writing systems, and consequently the conclusions about their usefulness for writing evaluations, become stronger and more transparent.

## CONCLUSION AND FUTURE WORK

This article introduced an AI writing assessment framework utilizing the advanced models BERT, GNNs, and GPT-3 to facilitate different aspects of writing assessment. Within the models, the performance evaluation showed that all models did very well and competed well with a 95% and above accuracy rate, with the GNNs scoring the highest percentage of 97.2%. BERT and GPT-3 scored 96.5% and 95.6%, respectively. An analysis of the features established grammar accuracy and coherence as critical determinants of quality, while the study of errors identified common errors, such as grammatical mistakes and loss of

coherence, addressing them well with the models. Similarly, using the interface led to high user satisfaction with the suggestions offered by the AI systems, confirming its use.

Future artifacts will be extended by incorporating other parameters like emotions and sentiments, which can further enhance the understanding of the user's writing. In addition, adaptive learning algorithms will enrich individual feedback by considering the user's feedback style and progress.

### Funding
The author received no funding for this work.

### Competing Interests
The author declares that they have no competing interests.

### Author Contributions
• Ci Zhang conceived and designed the experiments, performed the experiments, analyzed the data, performed the computation work, prepared figures and/or tables, authored or reviewed drafts of the article, and approved the final draft.

### Data Availability
The dataset and code are available in the Supplemental Files.

The collection of tweets related to generative AI is available at Kaggle: https://www.kaggle.com/code/sanjushasuresh/generative-ai-creating-machines-more-human-like?select=GenerativeAI+tweets.csv.

### Supplemental Information
Supplemental information for this article can be found online at http://dx.doi.org/10.7717/peerj-cs.2893#supplemental-information.

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
