# Peer review of "Optimising AI writing assessment using feedback and knowledge graph integration"

_PeerJ Computer Science, doi:10.7717/peerj-cs.2893_

## Round 0.1 · original submission · Major Revisions

Kindly revise the manuscript with particular attention to section related to the Methodology used, ensuring that the experimental design is clearly outlined and well-justified. Additionally, please provide a more detailed discussion of the validity of the findings, addressing any potential limitations or sources of bias in the experiments. This will help to enhance the robustness and credibility of the study.

·

Basic reporting

1. The article should ensure that all terms and concepts are clearly defined, especially those related to AI and knowledge graphs, to ensure that readers from diverse backgrounds can understand. Ensure that all key terms, especially technical ones, are clearly defined in the introduction or a glossary.

2. Include a broader range of literature to contextualize the study within the current research landscape.

Experimental design

1. The article could provide more detailed information on the experimental setup, including how the data was collected and the specific parameters used in the AI models, sample size justification, data collection methods, and model training parameters.

2. Reframe the research question to clearly outline the gap in knowledge this study addresses and how it contributes to the field.

Validity of the findings

1. The article should provide more information on the robustness of the data, including any potential biases in the dataset and how these were mitigated.

2. While the study provides some statistical analysis, it could be strengthened by including more detailed statistical tests and confidence intervals.

Additional comments

1. Some figures and images could be improved for clarity, particularly how they represent complex data or models.

Reviewer 2 ·

Basic reporting

The article presents an interesting aun up-to-date integration of advncements in the area of Machine Learning, ainmig to provide tailored assesments for " English as a Second Language" learners integrating Deep Learning systems, Knowledge Graphs, and Graph Neural Networks models.
Nevertheless, the readers may be confused, since in some parts of the text authors mantion that GPT -4 has been used, whereas in the theoretical description only GPT-3 is decribed . This should be modified.

Experimental design

The experiments themselves are promising, nevertheless the article lacks a pure Methodology section, describing gow different techniques, and corpora data interwine. I haven“t seen any description in the text, the profile of the 100 resposdents to the survey assessing the quality of the outcomes of this research.

Validity of the findings

The findings are meaningful.

Cite this review as

---

## Round 0.2 · accepted · Accept

Thank you for your revised submission. I confirm that all of the reviewers' comments have been addressed satisfactorily. While the original reviewers were not invited to re-review the manuscript, I have carefully assessed the revised version myself and am pleased with the quality and clarity of the current submission.

I am happy to confirm that your manuscript is now ready for publication.

Congratulations, and thank you for choosing to submit your work to PeerJ.